# Scanning Electron Microscopy Protocol for Studying Anatomy of Highly Degraded Waterlogged Archaeological Wood

Angela Balzano [ID], Maks Merela [ID] and Katarina Čufar *[ID]

Biotechnical Faculty, University of Ljubljana, SI-1000 Ljubljana, Slovenia; angela.balzano@bf.uni-lj.si (A.B.); maks.merela@bf.uni-lj.si (M.M.)
* Correspondence: katarina.cufar@bf.uni-lj.si

**Abstract:** Waterlogged archaeological wood (WAW), approximately 4500 years old, from the prehistoric pile-dwelling settlement at Ljubljansko barje, Slovenia, was examined by scanning electron microscopy (SEM). We propose a simplified protocol for sample preparation and the SEM technique for the study of highly degraded WAW of *Quercus*, *Faxinus*, *Acer*, *Salix* and *Populus*, representing taxa with different wood properties. We present the advantages of the proposed technique for wood identification, the observation of various anatomical features and for the study of cell wall degradation. SEM, equipped with energy-dispersive X-ray spectroscopy (EDX), allowed us to detect significant amounts of Fe, S and Ca with different appearances, amounts and distributions in the wood of the studied taxa. In the case of *Populus*, an increased amount of Si was also detected. The applied SEM protocol allowed characterisation of the anatomy of the highly degraded WAW while reducing the time required for sample preparation and examination under the microscope, as well as extending the lifetime of the SEM components (e.g., tungsten filament), compared to the situation when we analyse wood samples with a greater volume.

**Keywords:** waterlogged archaeological wood; scanning electron microscopy; wood anatomy; wood preservation; protocol; EDX

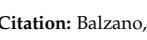



## 1. Introduction

Archaeological wood is an important source of information about the past. It is the main material source in dendroarchaeology, a sub-discipline of dendrochronology, whose original focus on providing absolute dating is now extended by the application of modern techniques that allow the reconstruction of past practices of wood selection and use, as well as the evolution of past environments and human societies [1,2].

Archaeological wood is mostly preserved in waterlogged environments where it becomes saturated with water and, consequently, anoxic or suboxic conditions are created [3]. Waterlogged archaeological wood (WAW), in environments with a low availability of oxygen, is over time invaded by fungi and bacteria that can degrade the cell walls. Among them, there are erosion bacteria that can be active under conditions of extremely low oxygen [4–8]. Erosion bacteria often coexist with other wood-degrading microorganisms, such as tunnelling bacteria and soft rot fungi [9].

When WAW is removed from its environment, it is difficult to prevent desiccation and preserve the material. Conservation is only possible if we understand the extent and cause of degradation [10]. Varying degrees of degradation can be visible at the microscopic level within the anatomical structure. Numerous characterisation methods have been used to determine chemical changes, as reviewed in [6–11]. The conventional methods of wood physics and chemistry have been extended by techniques, such as UV, FTIR and NMR [8]. In buried and waterlogged woods, hemicelluloses are usually degraded first, followed by cellulose and, finally, lignin [8]. Microscopy techniques, such as light and electron microscopy, help to characterise the degree and location of degradation at the cellular

and subcellular levels. Transmission electron microscopy (TEM) using potassium permanganate ($KMnO_4$) stained sections has helped to reveal cell wall degradation by erosion bacteria [5,12–15]. Erosion bacteria decompose secondary cell walls, including their highly lignified parts [6,10,15]. Several studies have shown that bacterial attack is microscopically detectable on the surface of the cell wall in the early stages of attack [16,17]. In advanced stages, the compound middle lamellae appear to be unaffected, while the secondary cell walls may be heavily degraded and appear as a granular mass likely consisting of residual lignin, bacterial slime and the remnants of erosion bacteria [18].

In Central Europe, large quantities of WAW come from prehistoric pile dwellings that existed around the Alps from about the fourth to the second millennium BC (e.g., [19–22]). Due to the long tradition of life on the lakes and marshlands, nearly 1000 archaeological pile-dwelling sites have been documented in six countries around the Alps, and 111 of them were inscribed on the UNESCO World Heritage List in 2011 [23].

Ljubljansko barje near Ljubljana, Slovenia, is the southeasternmost site of prehistoric pile dwellings around the Alps. WAW has been studied at 15 sites that existed between 3700 and 2400 BC [22–25]. WAW played a crucial role in dating the time of settlement and also in studying the paleoenvironment, forest management and other human activities like the integral use of trees and plants for food and everyday life (e.g., [26,27]). The preservation of WAW requires detailed characterisation potential, and therefore various studies have applied physical, chemical and microscopic techniques to characterise it [28,29]. All WAW from the Ljubljansko barje pile dwellings proved to be highly degraded [28]. These WAW had an increased lignin content, while the amount of cellulose and hemicelluloses was reduced [28,29]. Topochemical studies of the cell walls by transmission electron microscopy (TEM) and UV microspectrophotometry (UMSP) showed that the cell walls were severely degraded, while the middle lamella often appeared to be intact [29]. The changes in the cell wall were reflected in decreased density, increased moisture content, and shrinkage of the WAW [28]. However, careful handling of WAW after excavation has enabled it to be used in large studies, like those engaged in DNA sequencing, for which sufficient preservation of DNA is needed [30].

The development of various microscopy techniques has contributed significantly to the study of WAW [31]. Light microscopy (LM) is readily available, and therefore remains the most commonly used method to obtain information on the anatomy and chemistry of degraded wood. Scanning electron microscopy (SEM) has improved our knowledge of the cell wall structure and the various morphological forms of wood decay that were previously described by light microscopy [31,32].

SEM simultaneously allows viewing larger portions of the wood at lower magnifications or obtaining high resolution images of the three-dimensional wood microstructure. Under magnifications much larger than those of LM, SEM can better detect cell wall thinning caused by the loss of cell wall components [32,33], although much expertise is required to detect such changes because SEM does not allow staining (like LM) or contrasting (like TEM) of cell wall components. Thus, a combination of LM, SEM and other microscopy techniques is often used to provide complementary information from morphological to microstructural and chemical levels [29,31].

SEM can reveal traces of bacterial infestation, fungal spores, and cell wall detachment, which are crucial for understanding the mechanisms and processes involved in wood decomposition [31,33]. Depending on the research objective, various protocols have been developed to prepare samples of degraded wood for observation. SEM analysis must be performed in a vacuum and therefore requires drying the samples, which is the most difficult step in preparing WAW for SEM observation without causing damage, such as collapse and deformation. To prevent the wood from collapsing and cracking during drying, most protocols fix (e.g., with glutaraldehyde) the specimens before air drying or freeze drying. Such protocols are commonly used for the observation of sensitive features, such as bacteria, fungal spores, and hyphae [18,33,34]. In addition, less demanding SEM protocols can be used for documenting features, such as wood morphology. The latter

usually require cutting a small volume of wood with a razor blade, dehydrating the samples through an acetone series, and finally gold sputtering [35]. In addition, SEM, equipped with an energy-dispersive X-ray spectroscopy (EDX) system, allows the analysis of the inorganic chemical composition of the cell wall and various inclusions [32,36]. This technique thus offers the possibility of detecting and characterizing the minerals present in WAW, usually silica, pyrites or calcium carbonate, which indicate the beginning of the process of fossilisation [37].

The objective of the present study was to investigate the anatomy of highly degraded, waterlogged archaeological wood of various species from a pile-dwelling settlement that existed about 4500 years ago. To this end, we developed a scanning electron microscopy protocol (SEM) that allows simple but effective preparation of WAW for observation of large surfaces at low magnification, which is important for the study of wood anatomy and identification, while allowing observation of the progression of cell wall degradation at higher magnification and resolution. Here, we wanted to take advantage of energy dispersive X-ray spectroscopy (EDX) to detect the progress of wood mineralisation. Finally, we wanted to test how the proposed protocol helps to study the wood structure of species with different wood anatomy, properties, and states of preservation.

## 2. Materials and Methods

### 2.1. Waterlogged Archaeological Wood

The WAW originated from the Late Neolithic (Eneolithic, Copper Age) settlement of Špica in Ljubljana, Slovenia on the banks of the Ljubljanica River on the edge of Ljubljansko barje [38]. Ljubljansko barje is known for numerous prehistorical archaeological sites that were mainly located on the edge of the lake [22]. For the present study we selected the samples of WAW buried in the waterlogged ground for about 4500 years. The samples come from a larger collection of wood samples previously included in various studies of wood identification, dendrochronology, wood preservation and wood as a source of information on past environments and human activities. The samples, in the form of discs (approximately 10 cm in diameter and approximately 10 cm thick), were cut from the remains of the piles on which the prehistoric dwellings were built. We selected samples of oak (*Quercus* sp.), ash (*Fraxinus* sp.), maple (*Acer* sp.), poplar (*Populus* sp.) and willow (*Salix* sp.) which cannot be identified to the level of species. The selected taxa have wood with different properties in terms of density, mechanical properties, and durability, and were commonly used to construct buildings on the piles. The majority of the taxa, except oak, do not form coloured heartwood and their wood is not durable (European Standard EN 350-2, 1995 [39]). Since the WAW appeared to be uniformly preserved from the pith to the bark, the samples were taken from the outermost part of the collected discs. In the case of oak, the only species with durable heartwood [40], analyses were carried out on the outer heartwood, which appeared to be better preserved than the sapwood.

Selected WAW disks were divided into subsamples to be processed by different preparation methods for SEM.

The physical properties, maximum moisture content ($MC_{max}$) and density ($\rho_0$) of the WAW were determined, and light microscopy was used to additionally evaluate its degradation.

### 2.2. Physical Properties of WAW

The maximum moisture content and density of dry wood were used for additional characterisation of the wood, i.e., to assess the degree of wood degradation as proposed in previous research and English Heritage guidelines [28,41–43].

The maximum moisture content ($MC_{max}$) was gravimetrically determined on parallel samples (dimensions ca. $30 \times 30 \times 30$ mm) using Equation (1).

$$MC_{max} = [(m_{max} - m_0)/m_0] \times 100 \ [\%] \tag{1}$$

where:

$m_{max}$—mass of sample completely soaked with water;

$m_0$—mass of absolutely dry sample (drying in the oven at $103 \pm 2$ °C until constant mass is achieved).

The total water saturation of the samples (for $m_{max}$) was obtained by immersing them in water and applying a vacuum so that the trapped air was removed.

Dry density ($\rho_0$), based on the mass ($m_0$) per volume ($V_0$) of dry wood, was determined on twelve oriented parallel samples ($10 \times 10 \times 10$ mm) of each species. For this purpose, the samples were freeze-dried in a Telstar LyoQuest lyophiliser at 0.040 mbar and $-82$ °C [44]. The bulk volume and density of each sample were determined using the GeoPyc 1365 density analyser (Micromeritics Inc., Norcross, GA, USA) [45–47].

### 2.3. Scanning Electron Microscopy (SEM) and Energy Dispersive X-ray Spectroscopy (EDX)

After freezing the discs of WAW (Figure 1a,b) at ca. $-22$ °C, thin sections (dimensions around $5 \times 5$ mm and thickness of 20 µm) of WAW were cut by hand using a razor blade (Figure 1c). We separately prepared cross-, radial- and tangential sections. Slices were flattened on specimen mounts pre-treated with glycerine albumen (Agar Scientific Ltd., Essex, UK). Glycerine albumen, commonly called albumin, is already widely used in light microscopy for better adhesion of tissue sections to glass slides. One drop of albumin was spread on the mount surface, and the section was then placed on it (Figure 1c) and oven dried at 70 °C for 15 min to activate the adhesive properties of albumin. With this step we dried the sections without damaging them, and prevented the thin sections from collapsing and cracking in the SEM chamber.

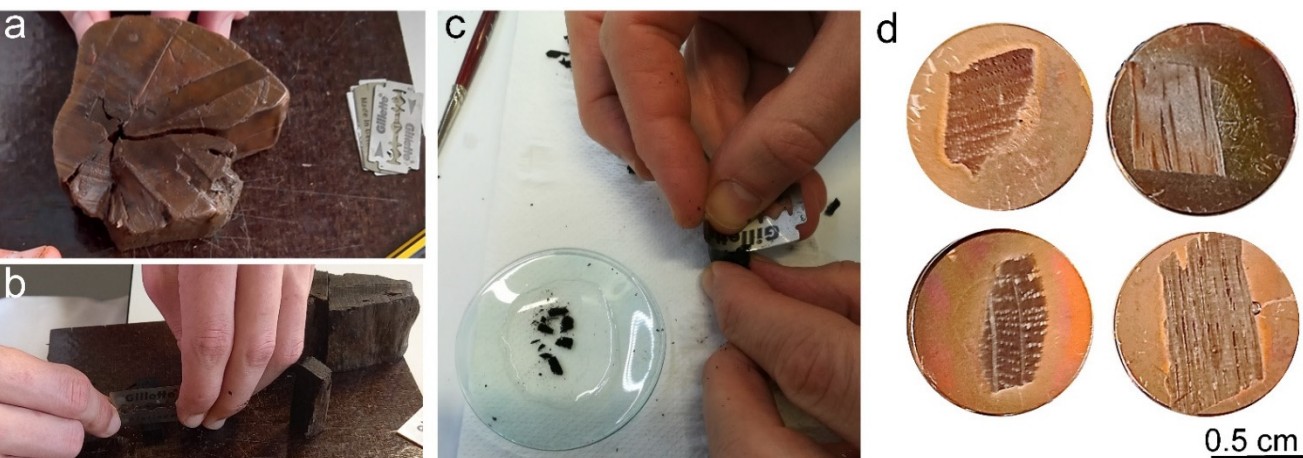

**Figure 1.** Preparation of thin slices of waterlogged archaeological wood (WAW) for SEM analysis. (**a**) Thin sections of wood were cut by hand from the frozen WAW disk using a razor blade (**b**,**c**) and were flattened on specimen mounts pre-treated with albumin (**d**).

The observation of the thin sections allowed us to work in high vacuum conditions and achieve high resolution at higher magnifications. At the same time, the sections were thick enough to provide a favourable depth of field and retain the advantage of observing the 3D structure of the anatomical features of the wood.

Due to the relatively small amount of material in the chamber compared to samples normally used for SEM analysis of wood [48] we were able to work in high vacuum, significantly reducing the time needed to reach a complete vacuum. Since wood is a porous material, the larger the samples, the more likely it is that inside the bulk there are pores which do not reach vacuum, but are still at atmospheric pressure even during observation under the electron beam. This can cause the oxidation of the tungsten filament and, consequently, the reduction of its lifetime. However, the lifetime of the filament can be prolonged by the use of thin sections.

The sections were subsequently coated with carbon (Q150R ES Coating System Quorum Technologies, Laughton, UK) for 30 s at 20 mA intensity. The carbon coating made the samples conductive without affecting the subsequent chemical characterisation.

Samples of recent wood of *Quercus* sp. and *Populus* sp. were prepared in the same way as for WAW and used as references. SEM micrographs were then taken under high vacuum and low voltage (between 5 and 15 kV) conditions. A large field detector (LFD) and a concentric backscatter detector (CBS) were used in a FEI Quanta 250 SEM microscope (Hillsboro, OR, USA). Observations were performed at a working distance between 8 and 10 mm.

Finally, energy-dispersive X-ray spectroscopy (EDX) was performed using a CBS detector at a voltage of 15 kV and a working distance 10 mm on selected points on each sample with different orientations. Analysis was made with a TEAM EDS analysis system (EDAX, AMETEK Inc., Berwyn, PA, USA) using the point analysis option. This method enabled us to determine the elemental composition at the points of interest in the sections observed with SEM.

### 2.4. Light Microscopy (LM)

Since light microscopy (LM) is commonly used to observe WAW, we used it here for comparison with SEM. We used two different methods of sample preparation, as each has some advantages and disadvantages.

The first method consisted of freezing discs of WAW at a temperature of ca. $-22\ °C$. Thin sections of wood were cut by hand from the frozen discs using a razor blade. We separately prepared cross-, radial- and tangential sections with dimensions of $5 \times 5$ mm and a thickness around 20 μm. The obtained microsections were mounted on glass slides and embedded in glycerine.

For the second method we obtained sub-samples ($5 \times 5 \times 5$ mm) from WAW disks for each anatomical plane, embedded them in paraffin using a Leica TP 1020-1 (Nussloch, Germany) and cut them with a semi-automatic rotary microtome RM 2245, Leica, (Nussloch, Germany). The sections (9 μm thick) were flattened on slides pre-treated with albumin and dried at 70 °C for 20 min, then cleaned of residual paraffin by washing with bioclear (Bio Optica, Milano, Italy) and ethanol. The sections were stained with a water solution of safranin and astra blue, and permanently mounted on glass slides in Euparal (Chroma 3C-239 Waldeck, Münster, Germany) [49]. All slides were observed under a Zeiss Axio Imager A.2 light microscope (Carl Zeiss Microscopy, White Plains, NY, USA), and images were acquired with a Zeiss Axiocam 712 colour (Carl Zeiss Microscopy GmbH, Jena, Germany). Samples of recent wood of *Quercus* sp. were prepared in the same way as for WAW and used as references.

### 3. Results and Discussion

### 3.1. Physical Properties of WAW

The physical properties of WAW, maximum moisture content ($MC_{max}$) and density ($\rho_0$) confirmed that the 4500-year-old archaeological wood of all the species studied was severely degraded, which is consistent with previous studies on WAW from nearby archaeological sites with a similar age, burial environment and history [28,29]. The $MC_{max}$ determined for the wood species studied ranged from 517 to 902% (Table 1). According to English Heritage guidelines [42], an $MC_{max}$ of >150% for oak or >400% for poplar means that the wood is highly degraded. For oak, in the area mainly represented by *Quercus petraea* and *Quercus robur*, normal recent wood would have an $MC_{max}$ value of about 100%, and similar values can be expected for ash and maple, while poplar and willow, due to their structure and low wood density, normally have an $MC_{max}$ value of over 200% [28].

**Table 1.** Physical properties of 4500-year-old waterlogged archaeological wood (WAW), maximum water content (MC$_{max}$) and dry density ($\rho_0$) of WAW compared to the oven-dry density ($\rho_0$) of normal wood from the literature [50,51].

| Wood Taxa | MC$_{max}$ (%) | WAW Density $\rho_0$ (kg/m$^3$) | Normal Wood Density $\rho_0$ (kg/m$^3$) |
|---|---|---|---|
| *Quercus*, oak (heartwood) | 747 | 180 | 650 |
| *Fraxinus*, ash | 517 | 140 | 650 |
| *Acer*, maple | 795 | 130 | 590 |
| *Populus*, poplar | 902 | 130 | 410 |
| *Salix*, willow | 653 | 140 | 400 |

The determined density of WAW varied between 130 and 140 kg/m$^3$ for archaeological ash, maple, poplar and willow; their normal sound wood would have average oven dry wood densities ($\rho_0$) of between 350 and 650 kg/m$^3$ [50,51]. The dry density ($\rho_0$) of archaeological oak was 180 kg/m$^3$, while its normal wood has an average density ($\rho_0$) of 650 kg/m$^3$. Thus, the density of WAW was about three to five times lower than that of normal wood, which confirms the strong degradation of archaeological wood [52].

*3.2. Scanning Electron Microscopy (SEM)*

The SEM protocol we used allowed us to make sections of frozen specimens with a razor blade, while fixation with albumin on specimen mounts prevented the occurrence of cracks and the collapse of large sections during drying, which is necessary for observation under SEM (Figure 2). In this way, we obtained high quality dry samples without using harmful chemicals, such as glutaraldehyde or acetone [18,33–35].

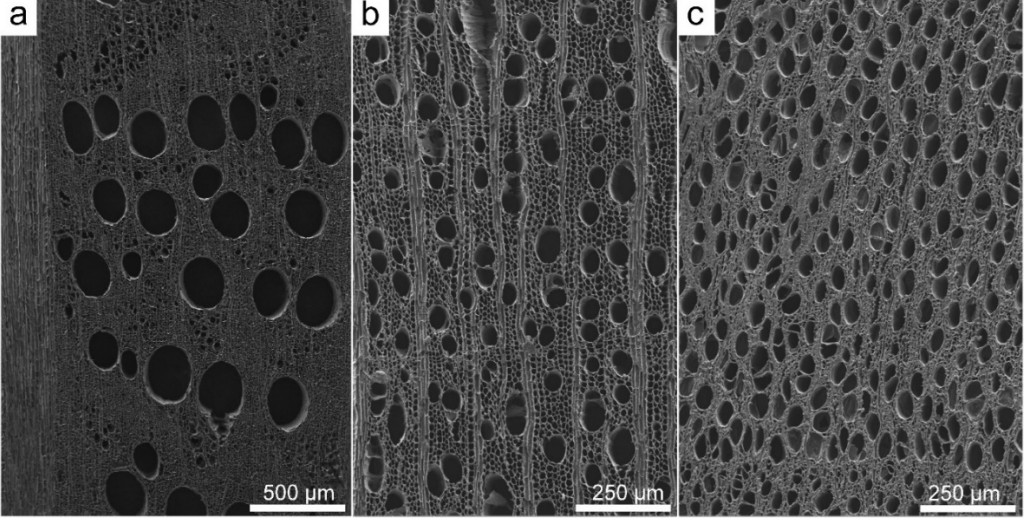

**Figure 2.** Large cross-sections of ca. 4500-year-old waterlogged archaeological wood of (**a**) oak (*Quercus* sp.), (**b**) maple (*Acer* sp.) and (**c**) poplar (*Populus* sp.).

At the same time, it was possible to obtain high quality photomicrographs of large parts of the wood at low magnification and of details at high magnification. The sections were thick enough to provide a favourable depth of field and retain the advantage of observing the 3D structure of the wood anatomical features (Figures 3 and 4).

High resolutions at higher magnifications, which vastly exceeded the magnifications and resolution of LM (Figures 2–4) were possible because working with thin sections allowed us to work in a high vacuum. Due to the small volume of the sections placed in the SEM chamber (dimensions of about 5 × 5 mm and a thickness of 20 µm, and volume 0.5–1 mm$^3$),

only a short time, about 2–3 min, was needed to create the complete vacuum, while it takes 10–15 min for usual wood samples with a volume of 1 cm³ [48]. We also avoided a problem associated with larger wood samples which contain more pores, as it is thus more likely that not all pores will reach the vacuum. This may lead to oxidation of the tungsten filament during the observation with the electron beam, thus shortening its lifetime. With the protocol proposed here, we could extend the lifetime of the filament by about 50 h.

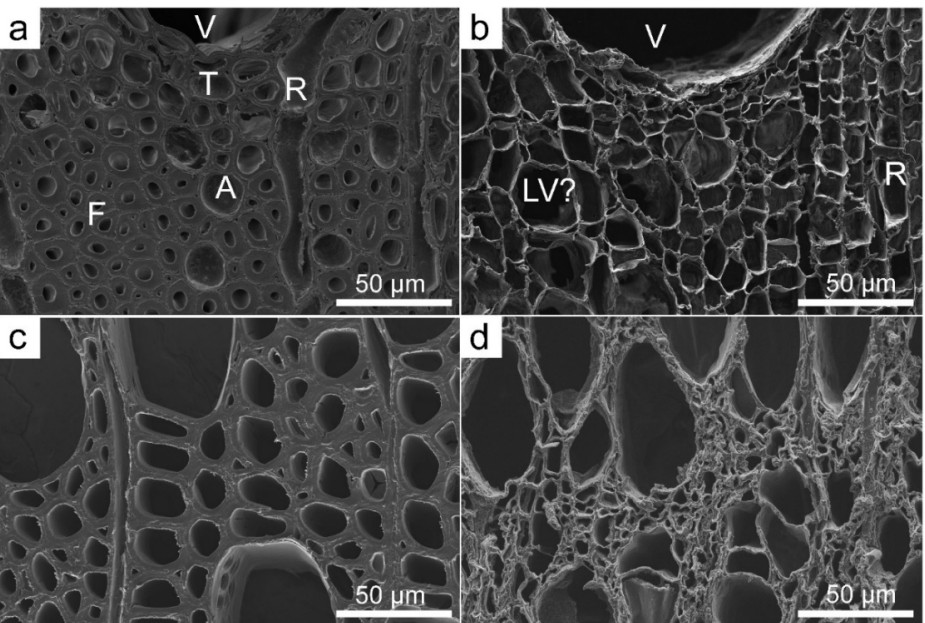

**Figure 3.** Cross-sections of sound recent wood of *Quercus* sp. (**a**) and *Populus* sp. (**c**) and corresponding waterlogged archaeological wood of the same species, respectively (**b**,**d**). F—fibre, T—vascular tracheid, A—axial parenchyma, V—vessel, LV—latewood vessel, R—ray.

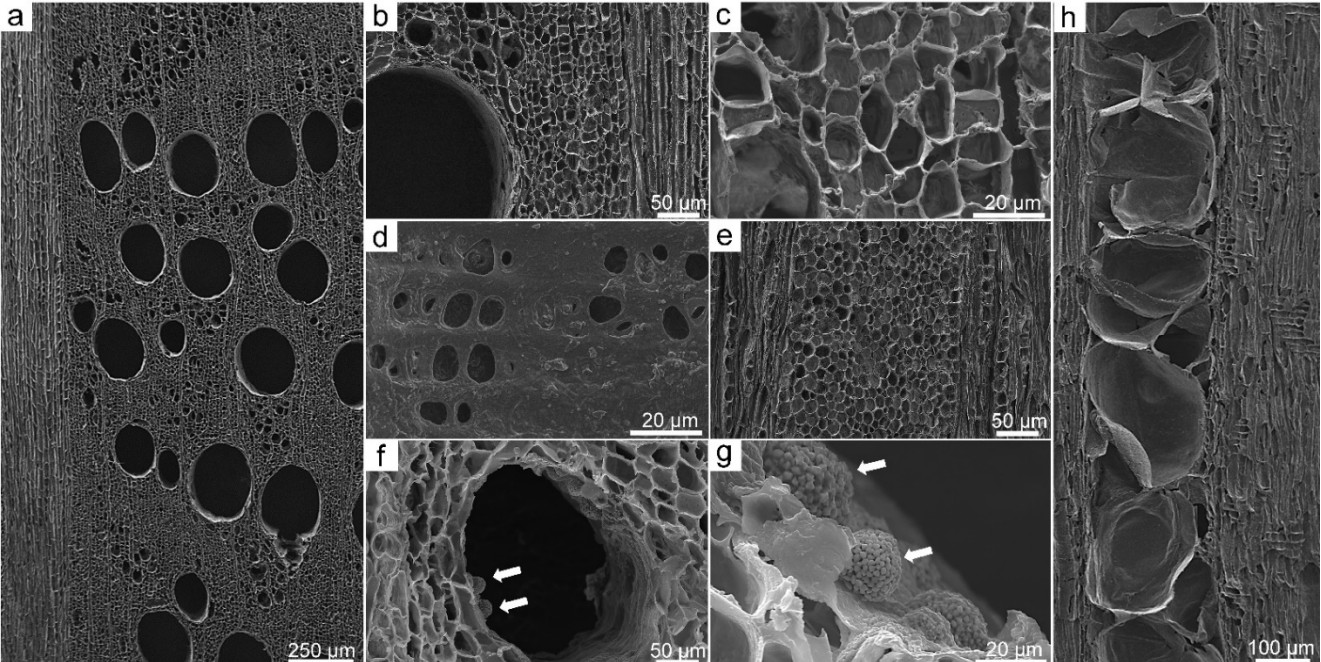

**Figure 4.** Micrographs of cross- (**a**–**c**,**f**,**g**) radial (**d**,**h**) and tangential (**e**) sections at different magnifications in *Quercus* sp. showing diagnostic features for wood identification and inclusions in the vessel (arrows) (**f**,**g**).

The samples for SEM are usually coated with a thin metal layer, usually gold, to become conductive [31]. In such cases SEM and EDX must be performed separately, since the metal may interfere with the chemical signal of the sample under investigation. The carbon coating in our study allowed us to obtain high quality images without any charging effect and presented the possibility of performing EDX directly, avoiding the use of expensive materials (i.e., gold).

The observations at magnifications of up to 8000× made it possible to view the small structures, like pits and their details, to evaluate the degradation and thinning of the cell wall, and to observe the surface of degraded cell walls with various deposits.

Observations at 5 kV in low vacuum with LFD proved to be optimal to observe the wood features [48], while inorganic inclusions could not be identified at this voltage and with this detector. Using 15 kV with a CBS detector in our study allowed observation at greater magnifications and detection of various inorganic inclusions. The best conditions to observe the inorganic inclusions were obtained at 15 kV with a CBS detector, which made it possible to obtain the chemical contrast (difference in composition, atomic number) (Figure 4g).

The observation under SEM at high magnifications showed the 3D structure and state of preservation of the cell walls, which varied with the species (Figures 3–8). The heartwood of *Quercus* sp. appeared to retain the structural features, although it was not possible to distinguish different cell types of similar size but usually different cell wall thickness, such as fibres, vascular tracheids and axial parenchyma, in cross-sections. These cells could not be differentiated as they all had very thin cell walls (Figure 3b). The WAW of *Populus* sp. generally appeared to be the most degraded of all species (Figures 3 and 7), and its wood appeared collapsed when observed under SEM (Figure 3d).

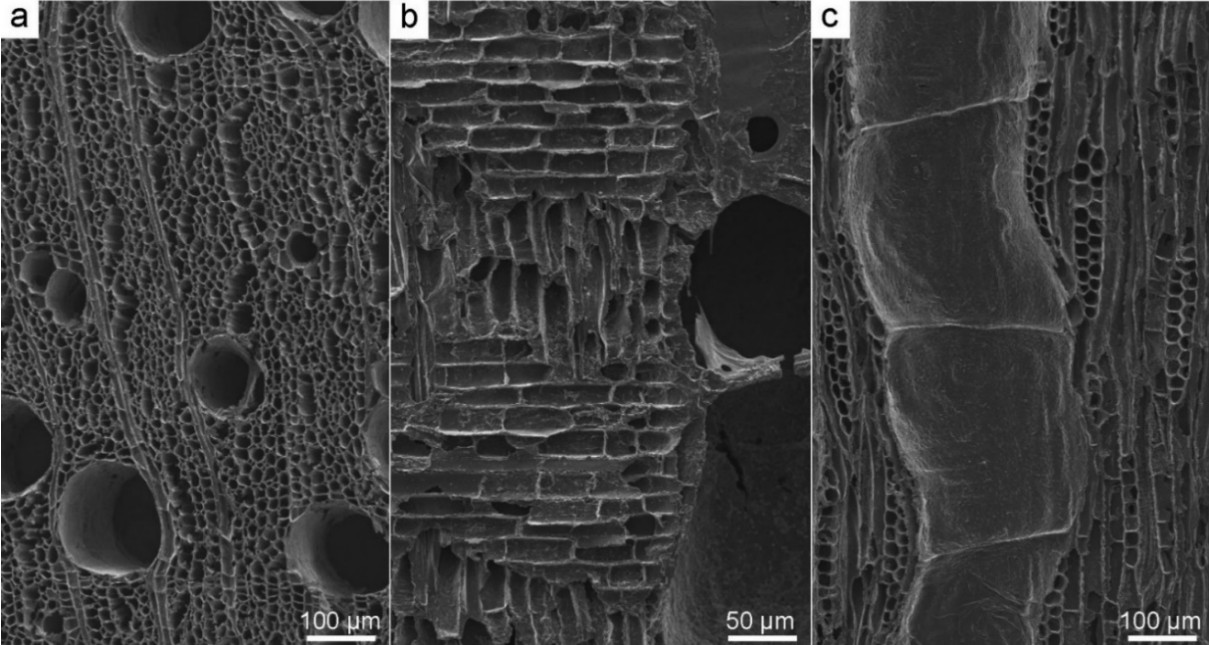

**Figure 5.** Micrographs of (**a**) cross-, (**b**) radial and (**c**) tangential sections of *Fraxinus* sp. at different magnifications showing diagnostic features for wood identification.

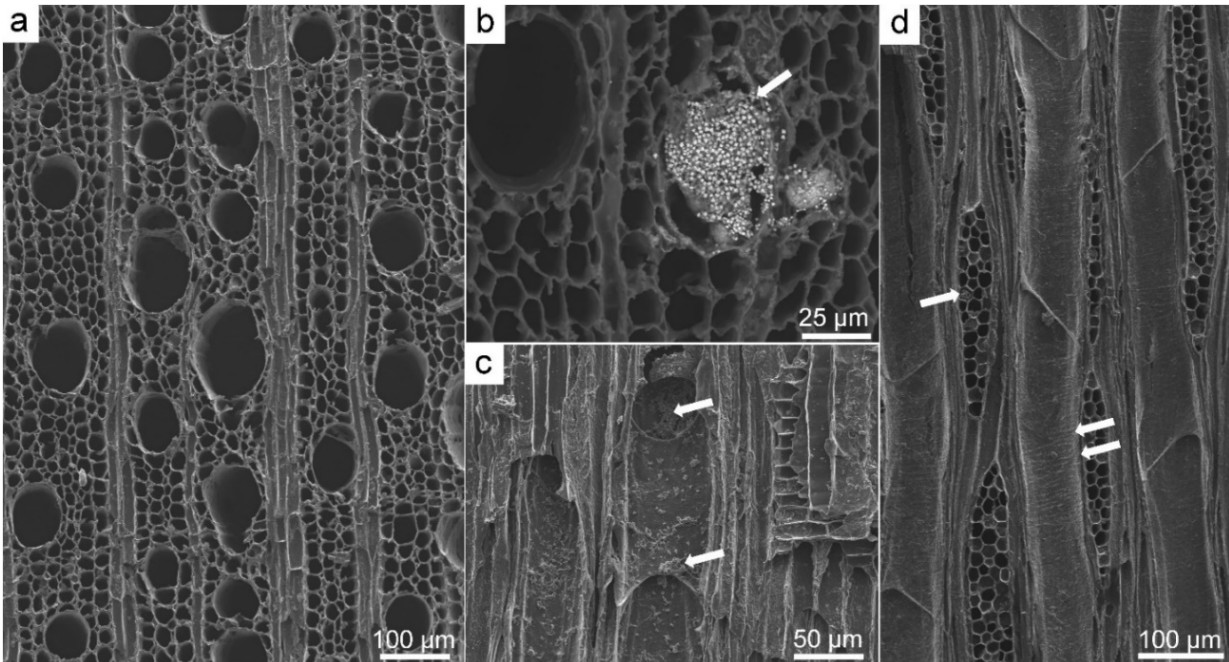

**Figure 6.** Micrographs of archaeological *Acer* sp. showing: cross-section of diffuse porous wood (**a**); inclusions in the vessel (arrow) (**b**); radial section with vessel elements and simple perforation plates, homogeneous ray tissue and numerous inclusions (arrow) (**c**); tangential section with homogenous rays containing inclusions (arrow) and vessel elements with spiral thickenings (double arrow) (**d**).

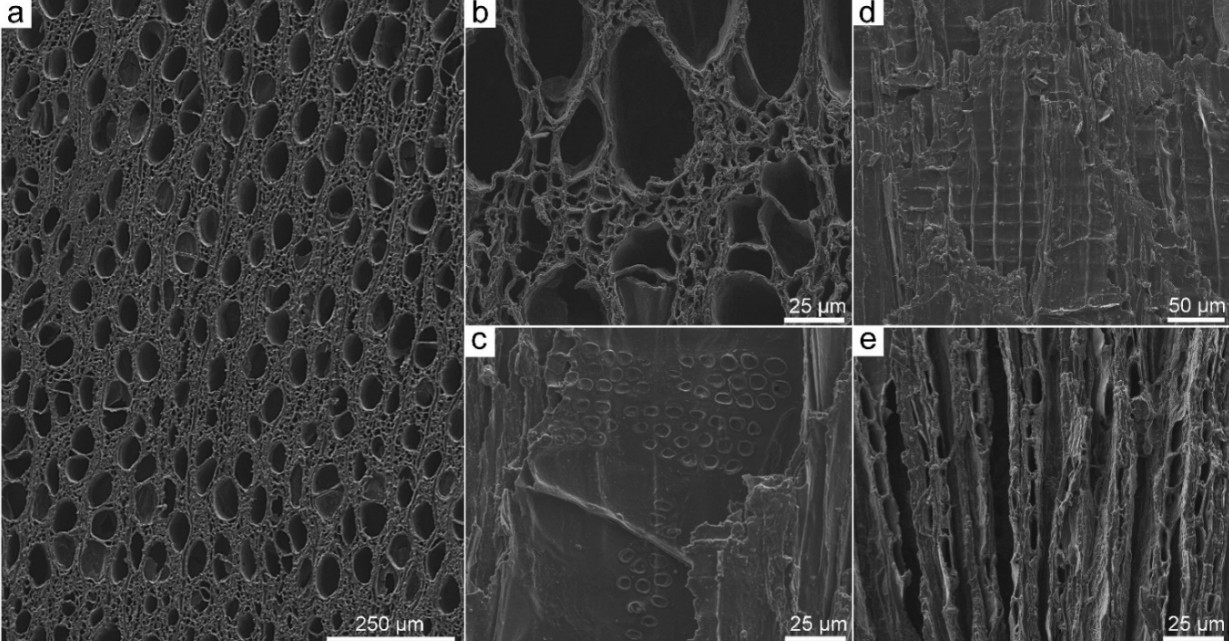

**Figure 7.** Micrographs of *Populus* sp.: cross-sections of diffuse porous wood (**a**) with cell collapse (**b**); radial section with large ray-vessel pits (**c**); radial (**d**) and tangential sections with uniseriate homogeneous rays (**e**).

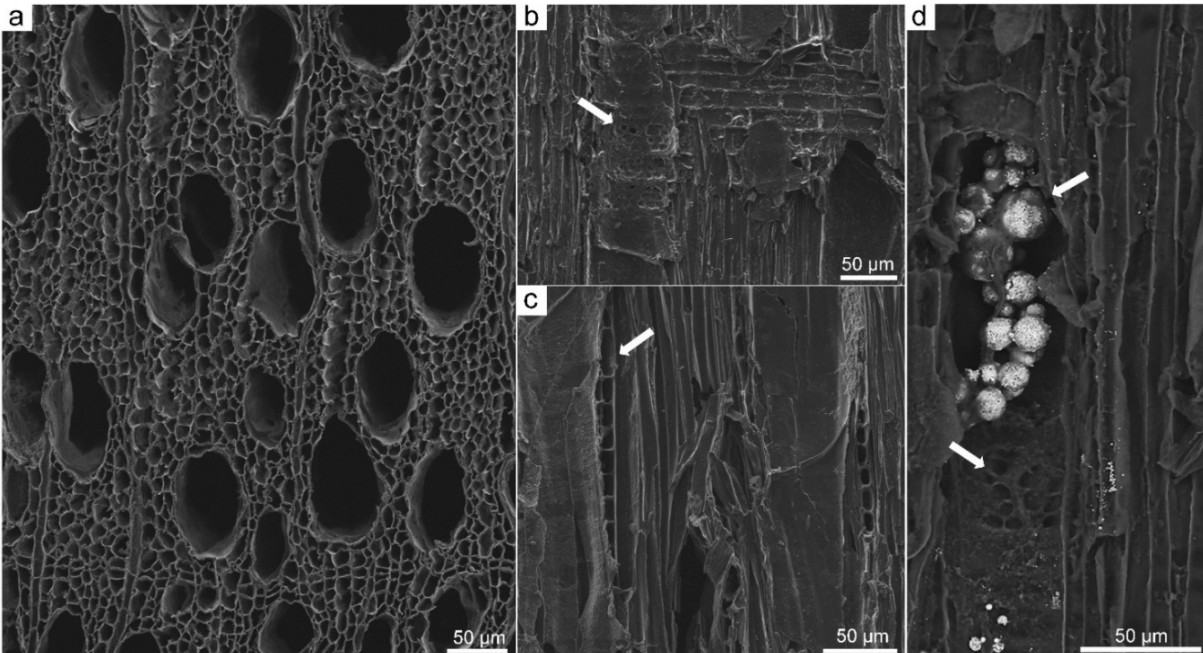

**Figure 8.** Micrographs of *Salix* sp.: cross-section with slightly collapsed vessels (**a**); radial section with large ray-vessel pits (arrows) (**b,d** lower arrow) and inclusions in the vessels (upper arrow) (**d**), radial section with heterogeneous rays and large ray-vessel pits in upright cells, and uniseriate heterogeneous rays with procumbent and upright (arrow) cells (**b,c**).

### 3.3. Application of SEM for Wood Anatomy and Wood Identification

SEM enabled the observation of the diagnostic wood anatomical features of WAW of five genera, *Quercus*, *Fraxinus*, *Acer*, *Populus* and *Salix*, which are mainly represented in the sampling region (prehistoric pile dwellings at Ljubljansko barje, Slovenia) by sessile (*Quercus petraea*) or pedunculate oak (*Quercus robur*), European ash (*Fraxinus excelsior*), narrow leaved ash (*Fraxinus angustifolia*) or manna ash (*Fraxinus ornus*), sycamore (*Acer pseudoplatanus*) and field maple (*Acer campestre*), black and white poplar or aspen (*Populus nigra*, *P. alba*, *P. tremula*), and white willow (*Salix alba*). The species within each genus could not be differentiated based on wood structure.

SEM images of oak showed a ring-porous distribution of earlywood vessels and latewood vessels arranged in a diagonal/radial pattern, as well as uniseriate and multiseriate rays (Figure 4). It was not possible to differentiate among vascular tracheids, fibres and axial parenchyma, as the cells were of comparable size and had thin cell walls due to bacterial degradation [8,15,29,53]. However, these cell types could be well differentiated on radial or tangential sections, based on the form, dimensions of cells and pits features. Tangential sections showed the size and distribution of rays and the condition of their cell walls. Tyloses in vessels were preserved and could be clearly seen in all sections.

Loss of cell wall material shown as thinner cell walls was the main sign of degradation in all cell types (Figure 4c). At greater magnifications it was possible to observe the changed surface of the cell walls towards the lumen and numerous deposits of various forms (Figure 4b,c,f,g).

Ash (*Fraxinus* sp.) is also ring porous (Figure 5a), while the latewood vessel distribution has no specific pattern. The fibres and axial parenchyma could not be differentiated on the cross-section, while radial and tangential sections enabled an insight into the structural details and signs of degradation, like generally thinner cell walls of vessels, fibres and axial parenchyma (Figure 5a). Simple perforation plates and homogeneous rays could be seen on the radial section (Figure 5b), while bi- to three-seriate rays could be observed on the tangential section (Figure 5c).

The wood of maple (Acer sp.) is diffusely porous, with small vessels uniformly distributed over the cross-section. Further diagnostic features are vessels with spiral thickenings, homocellular rays and characteristic vessel-ray pits [54]. In cross-sections fibres could not be differentiated from axial parenchyma, and growth ring boundaries were less distinct than in normal wood (Figure 6a,b). On longitudinal sections one could observe typical vessel elements, simple perforation plates and traces of spiral thickenings, while ray-vessel pits and their structure could also be observed (Figure 6c,d). Degradation of wood led to changes to the cell wall surface in vessels and numerous deposits in all cell types (Figure 6b).

Poplar (*Populus* sp.) (Figure 7) and willow (*Salix* sp.) (Figure 8) are both diffusely porous, and have nondurable wood with low density. The main diagnostic wood anatomical characteristics of poplar are a diffuse distribution of vessels, uniseriate homogeneous rays consisting of procumbent cells and large ray-vessel pits. The anatomy of willow (*Salix* sp.) is very similar, while the main features that distinguish it from poplar are uniseriate heterogenous rays composed of procumbent and upright cells.

WAW of poplar (*Populus* sp.) and willow (*Salix* sp.) tended to collapse during the preparation and microscopy stages, which affected the SEM observations.

### 3.4. Mineral Inclusions

SEM observation at 15 kV with a CBS detector revealed the presence of various inorganic inclusions (Figure 9) on cross-, radial- and tangential sections of all analysed species. *Quercus* sp. showed the lowest amount of mineral inclusions of all of the species, while *Populus* sp. contained the highest amount. Inclusions were mostly detected in the lumina of larger vessels, whereas in *Populus* sp. they were also found in smaller vessels and parenchyma cells.

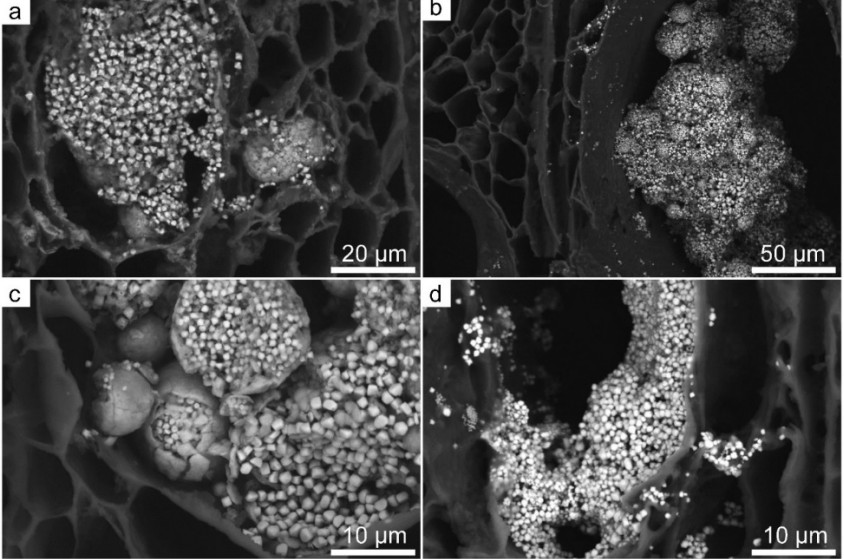

**Figure 9.** Mineral inorganic inclusions in WAW cross-sections of: *Acer* (**a**), *Fraxinus* (**b**)*, Salix* (**c**), and *Populus* (**d**) showing framboids (**a–c**) and individual crystals (**d**) under SEM with CBS at 15 kV.

EDX analysis of the inclusions revealed the presence of sulphur and iron (Figure 10a) in all species, while only *Populus* sp. contained abundant silicon (Figure 10b). Analyses also showed the presence of carbon and oxygen, which are the main elements in wood. Furthermore, other elements, such as calcium, magnesium, manganese, potassium, cobalt and sodium, were also detected. Crystals containing mainly iron and sulphur were found mostly in large vessels, while in *Populus* sp. these were also found in smaller vessels and parenchyma cells. Such crystals appeared as spheres (ca. 5–20 µm in diameter) made up of numerous smaller individual crystals (also smaller than 1 µm) (Figure 9c). The shape

of the inclusions, namely framboids, and their composition may indicate iron sulphides, like pyrite or marcasite [55,56].

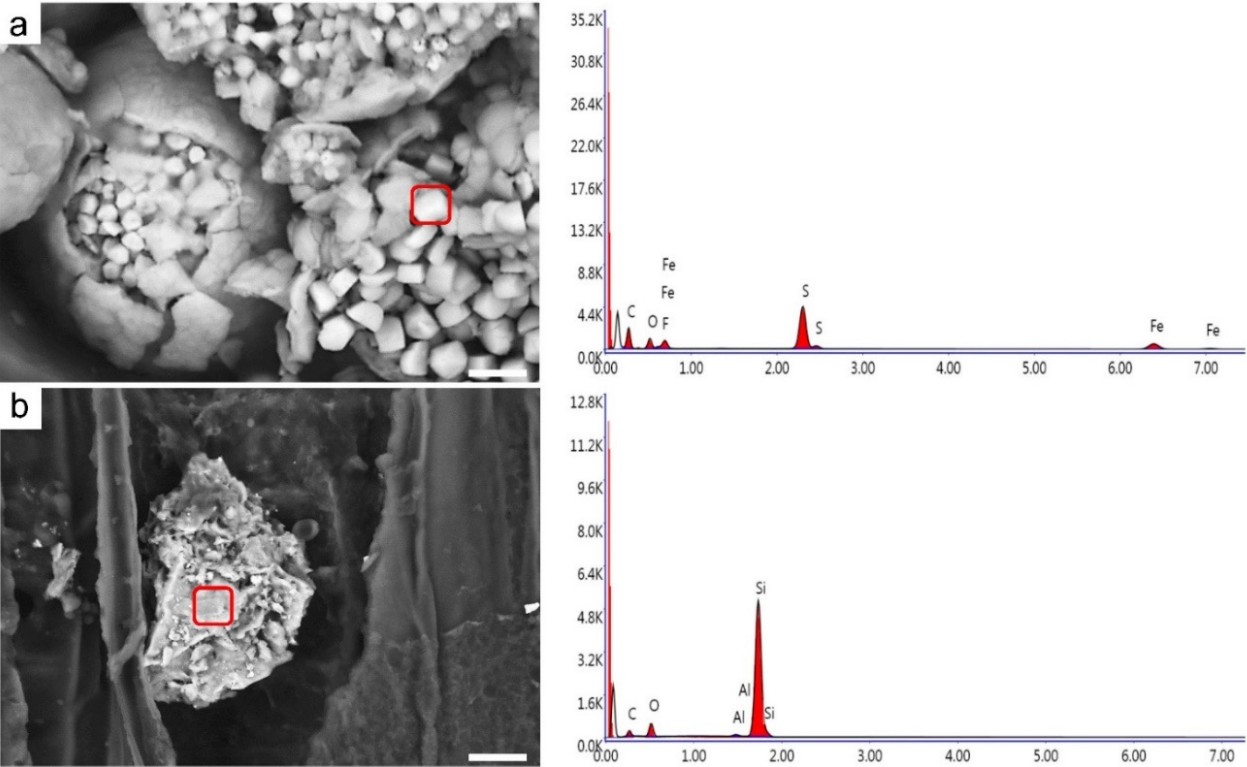

**Figure 10.** EDX point analyses of inorganic inclusions in *Salix* sp. (cross-section) and *Populus* sp. (tangential section) showing the differences in their compositions. Scale bar = 2.5 μm (**a**) and 10 μm (**b**).

Iron and sulphur are commonly found in waterlogged wood where sulphate reducing bacteria produce insoluble iron sulphide deposits [53,55,57–61]. Iron sulphide deposits themselves generally pose no threat to the preservation of WAW as long as the wood is water saturated, but they can cause damage under dry conditions, leading to the oxidation and incrustation of the WAW structure [60].

Silica inclusions (circa 20 μm in diameter) in *Populus* sp. appeared amorphous (Figure 10b). Their formation was reported to be associated with the presence of diatoms [61]. The massive presence of iron sulphides, like pyrite or marcasite, whose density (around 5000 kg/m$^3$) is considerably greater than that of the wood may have affected the results of density measurement (Table 1), especially in *Populus* which contained the greatest amount of crystals in the wood in our study.

The present results suggest that during the 4500 years burial history of the studied wood, the processes of gradual fossilisation (syn. petrification or permineralisation) had begun, accompanied by the accumulation of minerals, such as calcium carbonate (CaCO$_3$) or silicates (SiO$_2$), as reported in the literature [37]. Therefore, our results are consistent with other studies [31,32] on the usefulness and necessity of upgrading SEM with EDX to characterise the progress of mineralisation of WAW.

### 3.5. Light Microscopy (LM)

Light microscopy is a widely used method in the study of WAW. LM allows the identification of wood taxa, application of general and quantitative wood anatomy, and evaluation of degradation. For this purpose, it is important to consider how we prepare the sections, how we stain them, and what light mode and magnification we use.

Here we present LM for comparison and to illustrate some advantages of SEM, which was the focus of our study. For LM, we used two basic consolidation methods (freezing and paraffin embedding), two cutting methods (by hand and with a rotary microtome), which required different sizes of the original specimens and resulted in different surface sizes and thicknesses of the sections. We also used different treatments of the sections (unstained and stained) and different embeddings (glycerine or euparal) to obtain non-permanent or permanent slides.

The best results for WAW were obtained for cross-sections of 4500-year-old *Quercus* (Figure 11b,c,e,f), shown in comparison with images of recent wood (Figure 11a,d). We present here only the LM under bright-field, although it is known that polarisation or epifluorescence techniques can also be used successfully to enhance the details [29,62–66].

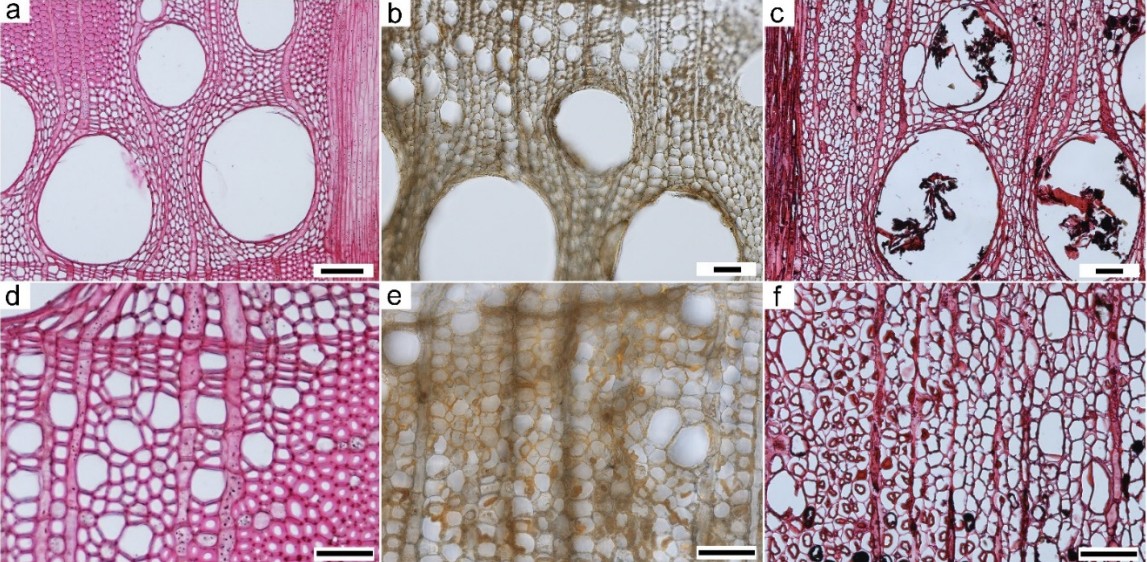

**Figure 11.** Light microscopy of *Quercus* sp. cross-sections at low (**a–c**) and high (**d–f**) magnifications. Recent wood (**a,d**); hand cut sections from frozen WAW, thickness >20 μm (**d,e**) and samples embedded in paraffin and cut with a rotary microtome, thickness 9 μm (**c,f**). Scale bar = 100 μm (**a**), 50 μm (**b–f**).

The method using thicker and larger hand-cut sections (Figure 11b,e) allowed a large portion of the cross-section to be viewed and the major cellular and tissue level structures important for wood identification to be seen, but due to the thickness limitation (20 μm) it was not possible to use the higher magnifications required to observe minute details, such as pits and pit structure. By embedding in paraffin (Figure 11c,f) and obtaining smaller and thinner sections, we were able to better observe the preservation of cell walls at higher magnification (Figure 11a,d), but the thin sections tended to tear due to the low fragility of the degraded cell walls.

We show here only cross-sections of *Quercus* (Figure 11), as the methods used did not allow us to make high quality images of other anatomical sections and other species studied, some of which were presented in a parallel study [66].

For *Salix* and *Populus* in particular, the poor state of preservation of the wood made it impossible to prepare slides of adequate quality using the same methods as for *Quercus*. The wood of *Salix* and *Populus* appeared to be torn and collapsed, making it impossible to obtain adequate images to identify the wood anatomical features. SEM was distinctly superior for this purpose (Figures 7 and 8).

Due to the poor quality of the slides, we were unable to demonstrate the other advantages of LM, such as the use of polarised light to determine the loss of cellulose [62], or the use of epifluorescence and an appropriate combination of stains to determine the presence and distribution of lignin [29,63].

Although SEM performed better than LM, the two techniques should be combined with each other and with other techniques (e.g., [67]) to provide complementary information from the morphological to the microstructural and chemical levels of wood structure.

## 4. Conclusions

We present a simplified scanning electron microscopy protocol for the observation of highly degraded, waterlogged archaeological wood.

The proposed SEM protocol consists of freezing the WAW and cutting sections with a razor blade. The transverse, radial, and tangential sections were adhered to specimen mounts with albumin, and dried at 70 °C for 15 min.

The use of albumin to fix the specimens to the holders prevented the occurrence of cracks and collapse of approximately $5 \times 5$ mm $\times 20$ µm large sections during drying without using harmful chemicals normally used for fixation and dehydration of specimens for SEM.

Due to the small volume (0.5–1 mm$^3$) of the sections placed in the SEM chamber, only about 2–3 min were required to create a complete vacuum. The vacuum reached the entire volume of the sample, preventing oxidation of the tungsten filament and extending its life by about 50 h. Unlike conventional protocols that sputter gold, we coated the samples with carbon only. This allowed us to obtain high-quality images and perform EDX analyses directly during microscopy, avoiding the use of expensive materials, such as gold.

The protocol made it possible to obtain high-quality micrographs of large portions of the wood at low magnification and of details at high magnification (up to 8000×) that far exceeded the magnifications and resolution achievable by optical microscopes. The sections were also thick enough to provide a high depth of field and preserve the advantage of observing the 3D anatomy of the wood.

Furthermore, it was possible to image small structures, such as pits and their details, traces of helical thickenings thinning of the cell walls, and various deposits.

SEM combined with EDX analysis of the inclusions revealed elevated amounts of sulphur and iron in all species, while only *Populus* contained abundant silicon, presumably indicating an early stage of fossilisation.

The study shows that the SEM protocol developed allowed the preparation of specimens to obtain high quality and high-resolution images for the study of wood anatomy under different magnifications. High quality images could be obtained for all anatomical sections (transverse, radial and tangential) for all taxa *Quercus*, *Faxinus*, *Acer*, *Salix* and *Populus* with different wood anatomy and very poor preservation, as also shown by physical testing, where the density was about three to five times lower than that of normal wood.

Although we presented numerous advantages of SEM compared to LM, the two techniques should be combined with each other and with other techniques to obtain complementary information, from the morphological to the microstructural and chemical levels of wood structure.

**Author Contributions:** Conceptualisation, A.B. and K.Č.; methodology, A.B., M.M. and K.Č.; validation, K.Č. and A.B.; formal analysis and investigation, A.B. and M.M.; data curation, K.Č.; writing—original draft preparation, A.B. and K.Č.; writing—review and editing, A.B. and K.Č.; visualisation, A.B. and K.Č. All authors have read and agreed to the published version of the manuscript.

**Funding:** The study was supported by the Slovenian Research Agency (ARRS), program P4-0015.

**Acknowledgments:** We thank Luka Krže, Davor Kržišnik and Andraž Benedik for their help with laboratory work, Željko Gorišek and Aleš Straže for comments on physical properties of wood, Miha Humar for providing the laboratory instruments at the Department of Wood Science and Technology, Paul Steed for editing the English, as well anonymous reviewers for their valuable comments and suggestions for improving the manuscript.

**Conflicts of Interest:** The authors declare no conflict of interest.

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
