# Peer review of "Scanning Electron Microscopy Protocol for Studying Anatomy of Highly Degraded Waterlogged Archaeological Wood"

_forests, doi:10.3390/f13020161_

Round 1

Reviewer 1 Report

Dear Authors

This work mainly documented the preservation state of 4,500-year-old piles recovered from a prehistoric dwelling settlement in Slovenia. Nonetheless the potential of the proposed “improved” SEM protocol is considered that hasn’t been adequately demonstrated and mostly it hasn’t’ been evaluated in comparison to any other SEM protocol. However, it is believed that overall this work it is a very interesting case study on the preservation state of prehistoric piles and therefore can be a useful contribution to the field of waterlogged archaeological wood if the document is revised and re-worked.

Title

The title of the paper is not considered representative of the work presented as authors mostly assessed the preservation state of WAW based on density, moisture content, morphology and inorganic inclusions (EDS). Two different protocols on light microscopy and one “new” on SEM were applied to document wood anatomy. As mentioned earlier the potential of the proposed SEM protocol to document severe decay, hasn’t been demonstrated and it has been comparatively evaluated.

Therefore, it is recommended to adjust the title, in order to be more representative of the content (see some examples below) or to run this SEM protocol in comparison to a well-accepted SEM protocol on highly degraded samples in order to show cell morphology and degradation patterns and prove its actual pros and cons.

Examples

“Documenting the cellular morphology of 4,500-year-old piles recovered from a prehistoric dwelling settlement”

“Morphological assessment of the degradation of waterlogged wood recovered from a prehistoric dwelling settlement in Slovenia”

“Microscopy studies of waterlogged archaeological wood retrieved from a prehistoric dwelling settlement”

Introduction

Some background info mentioned in the introduction, is not scientifically accurate and some citations are not appropriately used.  

Line 30-31

Saturation with water (waterlogging), doesn’t not normally create anoxic conditions. Do you mean disoxic? suboxic?

Please See Jordan et al. 2001, DOI: 10.1016/S0964-8305(01)00035-X)

Line 32-34

In low oxygen conditions wood microbiota includes fungi and bacteria which principally attacks cell wall polysaccharides. Moreover, erosion bacteria, even though they have the capacity, they don’t typically attack lignin

Please See Pedersen et al 2014,  DOI: 10.1515/hf-2013-0228

Line 70

Intact middle lamella doesn't necessarily mean that wood has been attacked by erosion bacteria. There are several microorganisms that preferentially degrade polysaccharides. Moreover, the reference used [28] doesn't support the presence of erosion bacteria. In contrast these researchers have proposed that wood was degraded by abiotic rather than biotic mechanisms. Even though I disagree with their hypothesis, this citation cannot be used in this sentence

Materials and Methods

Lines 100-101

Waterlogged archaeological wood can exist only in waterlogged soils. Please clarify in the burial/excavation environment was wet or waterlogged because this is considered important concerning the %MC measurements followed.

Lines 100-101

Water content. Do you mean Moisture Content?

Actual moisture content (MC %), of waterlogged wood is typically the weight of water in the waterlogged sample (Ww) expressed as a percentage of the oven dry weight (Wod), MC%= (Ww–Wod)/ Wod x 100 (Kollmann and Côté 1968, Skaar 1988 etc).

However, for the determination of maximum moisture content (MCmax) samples should be first vacuum impregnated in water to expel any trapped air and then weighted for recording their maximum waterlogged weight (Smith, 1954; Grattan 1987; Schniewind, 1990, McConnachie et al 2008).

The MC and MCmax are equal, only if samples are fully waterlogged.

So, were the samples fully waterlogged? (wet or waterlogged site?) or you vacuum impregnate them in water?

Please See

Smith (1954). Maximum moisture content method for determining specific gravity of small wood samples.

Schniewind, A. P. (1990). Physical and mechanical properties of archaeological wood. In Rowell, R. M., & Barbour, R. J. (Eds.). Archaeological wood: properties, chemistry, and preservation. American Chemical Society.

McConnachie, G., Eaton, R., & Jones, M. (2008). A re-evaluation of the use of maximum moisture content data for assessing the condition of waterlogged archaeological wood. E-preservation Science, 5, 29-35.

Lines 123-124

EN 13183-1, is not a recommended standard for waterlogged archaeological wood as it applies to sawn round timber, and timber which has been planed or surfaced by other means. This standard uses a 2cm test slice taken from a specific position of the swan timber, possibly for avoiding bias due to sawing.

This is not the typical process followed for wood,  please see classic standards i,e ASTM D 4442 , ISO-3130-1975, or books i.e. Kollmann and Côté 1968, Skaar 1988 etc

The standard used, EN 13183-1, is considered that it doesn’t comply neither with the recommended processes for the determination of the max moisture content (see Smith 1954; Schniewind, 1990, McConnachie et al 2008) nor to MC typical procedures applied on waterlogged archaeological wood (i.e. Grattan 1987, Florian 1990, Schniewind, 1990). 

Grattan, D. W. (1987). Waterlogged wood. In Conservation of marine archaeological objects (pp. 55-67). Butterworth-Heinemann.

Florian, M. L. E. (1990). Scope and history of archaeological wood. Advances in Chemistry Series 1990 No. 225 (pp.1-32)

Lines 129-130

Skeletal volume and density are mentioned in Materials and Methods but not in Results

Lines 153-179

For the SEM protocol several issues are considered that require clarifications.

Info on albumin are not provided (mode of application, concentration, manufacturer etc). Is this material a type of adhesive made from serum albumen ? bovine serum albumin ? chicken albumen? Is it conductive?

Elaboration is also needed on what authors mean by “slices were less prone to collapse and cracking due to drying during microscopy”? In all high vacuum SEM protocols, samples must be completely dried before are placed in the column. So, it is not clear how a completely dried sample will further shrink. Moreover, samples used in this study were dried at 70 oC.

Reduced time to reach the vacuum is stated as an advantage for this protocol due to samples’ smaller volume. However, a cubic sample could have less volume than a thin section, thus vacuum time is not shape-dependent, but volume-dependent. As the surface area of the thin sections is not provided, it is not very clear how this protocol reduces vacuum time compare to typical protocol for WAW were samples are commonly ~3x3x3 or ~4x4x4. Moreover, in several cases, cubic samples are required to be mounted to view two surfaces at once, or to be tilted at specific angles. Can thin sections provide relevant info?

Carbon cord coating is not recommended for SEM of delicate features such as wood deteriogens or degradation patterns and morphology at a cellular lever. Charging is a major problem which occurs due to wood non-conductivity and high porosity. In addition, with carbon coating wood specimens are sensitive to beam damage, especially at high magnifications. Therefore, if EDS analysis is needed, after EDS samples are usually sputtered with gold. Why this option was excluded.

The section on EDS is considered that doesn’t belong to microscopy. It is recommended to be a separate section of M&M or be a part of the preservation state assessment (2.2) as EDS is an independent analysis of the inorganic elementals of the material and is not related to wood morphological microscopic features

Results

Comparison of the envelop density to oven-dry density, is not considered scientifically appropriate. Envelope density should have been measured on sound wood of the same species. Otherwise basic density (green volume, dry weight) should be used for sound wood instead of oven-dry density, as basic density is the most commonly density used in waterlogged wood because it uses the green volume which is virtually the same as the waterlogged volume (Grattan 1990). Therefore, it is recommended to provide values obtained on sound wood samples by using the same instrument or provide basic density values for sound wood from the literature.

Moreover, calculation procedure for oven dry density of waterlogged wood (Lines 186-197) hasn't been mentioned in Materials and methods. It is considered important to explain how the dry volume was measured as waterlogged wood tend to shrink and disintegrate during oven-drying

The section on the use of SEM for wood ID is considered out of the scope the paper. The pros and cons of light microscopy versus SEM or CLSM etc are well known for wood ID. Wood identification keys worldwide are based on light microscopic features to ensure that wood anatomists around the world adopt a uniform protocol for recognizing and describing features, (IAWA lists, Schweingruber, 1978, Dallwitz, et al. 1995 (Delta-Intkey), Wheeler 2011 (Inside Wood) etc).  SEM , CLSM etc are used only where permanent or temporary thin sectioning is not feasible

Please see Cartwright, C. R. (2015).  doi: 10.1093/aob/mcv056

Line 341

It is stated that framboids, are associated to the activity of sulphate reducing bacteria however many studies report formation of pyrite framboids under strictly abiotic conditions questioning their biogenic origin

Please see Duverger et al 2020 doi : 10.3389/feart.2020.588310

Conclusions

Combination of SEM with EDS for wood analysis is well recognized since the 70s (Olsen 1983, Collet 1970 etc) and for deteriorated or waterlogged wood is also well-known (Daniels 2016 and references therein, and High and Penkman 2020). So, it is considered that nothing really novel has been stated with this conclusion.

Daniel, G. (2016). Microscope techniques for understanding wood cell structure and biodegradation. In Secondary xylem biology (pp. 309-343). Academic Press.

High, K. E., & Penkman, K. E. (2020). A review of analytical methods for assessing preservation in waterlogged archaeological wood and their application in practice. Heritage Science, 8(1), 1-33.

It is also stated that this protocol enabled analysis of samples without cracking and collapsing during microscopy. Are there SEM established protocol, where samples collapse during microscopy? Freeze sectioning/ fracturing, fixation, CDP (or FD) and cold sputtering may provide excellent results without cracking and collapsing. Even solvent exchange (graded series) process is a cheap and easy way that can also provide very nice results even for highly degraded wood.

The statement that the inclusions found in the wood indicate bacterial degradation of wood, cannot be supported by the obtained results and it is considered arbitrary. Which inorganic constituents are indicative of bacterial attack? Do you mean that sulfur reducing bacteria decay wood? Please clarify with relevant citations

Moreover, conclusion on the speed of analysis is also questionable as time is related to the surface area examined not to the shape or to the preparation protocol adopted.  A 2D sample could have larger surface area than a 3D one.

Finally, the filament life is also considered that is not related to the time needed to achieve vacuum or to the protocol, but to the size of the surface area examined regardless of its shape (2D or 3D)

Beside these comments, remarks and suggestions are annotated on the pdf of the revised manuscript.

Reviewer 2 Report

By freezing, good preparations for light microscopy would also be possible, which would also provide a large part of the results obtained. However, those who have the possibility to use an electron microscope should try and apply the proposed method.
When freezing the WAW, the formation of ice crystals and the associated spatial expansion also causes stresses in the wood, which can lead to cell collapses, especially in very decomposed woods. Freezing in water would be more gentle, as the tensions in the cell walls are reduced by the "water bond". Whether and how the wood can then be prepared for electron microscopy would be a future aspect of investigation.

Author Response

Thank you very much for your comments. Your suggestions provide a good opportunity for future studies.

Reviewer 3 Report

Angela et al. present an improved protocol for the identification of several types of waterlogged archaeological wood with SEM. The study is clear. And, the methodology is straight forward. The manuscript structure is good, English language is mostly well used except for some formatting error. However, as a methodology study, more details in the method section are encouraged to be provided in the manuscript. The following are a few suggestions:

Page 3, line 134 & Page 4, line 153:

There are two identical section numbers. Please check.

Page 3, line 145-146:

Please provide more details of the albumin treatment, i.e., treatment time and dose of the albumin solution.

Page 4, line 156:

The authors should describe how did they filter the slices with certain dimensions (500 x 500 μm and thickness of 20 μm as described in the manuscript), considering the extremely small size of the slices and this was performed manually by a razor blade not rotary microtome. Was the filter performed under stereo microscope?

Page 4, line 157:

Again, please provide more details of the albumin treatment. Was this pre-treatment preformed as described in Page 3 line 145?

Page 4, line 165-166:

There is an improper return in here. Please revise the format.

Page 6 line 232-233:

Please specify the number or the range of “a couple of minutes”.

Page 7, line 225 & Page 8, line 261:

There are two identical section numbers. Please check.

Page 7-10:

Please Italic the Latin names of the plants throughout the manuscript. Including but not limited to line 251, 254, 258, 263, 265, 266, 267, 285, 307, 310 and 313.

Author Response

Rev3

Angela et al. present an improved protocol for the identification of several types of waterlogged archaeological wood with SEM. The study is clear. And, the methodology is straight forward. The manuscript structure is good, English language is mostly well used except for some formatting error. However, as a methodology study, more details in the method section are encouraged to be provided in the manuscript. The following are a few suggestions:

Reply

Thank you for the comments. We are glad you for appreciate our research. We added more details regarding the method as required. We hope you will find our manuscript improved.

Rev3

Page 3, line 134 & Page 4, line 153:

There are two identical section numbers. Please check.

Reply: Thank you. We corrected this.

Rev3

Page 3, line 145-146:

Please provide more details of the albumin treatment, i.e., treatment time and dose of the albumin solution.

Reply: We added the requested information.

Rev3

Page 4, line 156:

The authors should describe how did they filter the slices with certain dimensions (500 x 500 μm and thickness of 20 μm as described in the manuscript), considering the extremely small size of the slices and this was performed manually by a razor blade not rotary microtome. Was the filter performed under stereo microscope?

Reply: Sorry we made a mistake. The dimensions were (5 x 5 mm and thickness of 20 μm).

Yes, the cutting was performed by hand with a razor blade and naked eyes.

Rev3

Page 4, line 157:

Again, please provide more details of the albumin treatment. Was this pre-treatment preformed as described in Page 3 line 145?

Reply: In lines 125-130 we provided the information.

Rev3

Page 4, line 165-166:

There is an improper return in here. Please revise the format.

Reply: We revised it.

Rev3

Page 6 line 232-233:

Please specify the number or the range of “a couple of minutes”.

Reply: We specified the range, it is 2-3 minutes.

Rev3

Page 7, line 225 & Page 8, line 261:

There are two identical section numbers. Please check.

Reply: We corrected this.

Rev3

Page 7-10:

Please Italic the Latin names of the plants throughout the manuscript. Including but not limited to line 251, 254, 258, 263, 265, 266, 267, 285, 307, 310 and 313.

Reply: Thank you. Yes, we corrected that.

Round 2

Reviewer 1 Report

Dear Authors

After the revisions and clarifications provided, it is believed that the manuscript is greatly improved and it is much clearer now that this work mostly aimed to introduce an alternative protocol for scanning electron microscopy which would allow wood species identification and concurrently would permit a condition assessment of the waterlogged material.

However, it is believed that several issues still need to be addressed so that the manuscript will be scientifically sound and significant of content.

Title

The title of the paper is still not considered representative of the work.

The results presented don’t support that this protocol can show degradation patterns whatsoever. SEM micrographs showed, of low magnification (< x3000 – x5000 ), didn’t demonstrate any fungal or bacterial degradation patterns of severely decayed wood.

In contrast it was well showed that this protocol could be useful for wood identification of highly degraded material.

Therefore, the title should be revised in order to represent the actual content of the paper and at the same time to show originality/ novelty and promote this alternative SEM protocol

Some examples

Wood identification of highly degraded waterlogged archaeological wood via a new SEM protocol

An alternative SEM protocol for identifying highly degraded waterlogged archaeological wood

An SEM protocol for species identification of highly degraded waterlogged archaeological wood

Introduction

It is considered that introduction should highlight why this work is important. The current state of microscopy of WAW could be reviewed and cited in order to show the purpose of the work and its significance. If current protocols of SEM and LM have flaws or cons this needs to be better highlighted in order to justify why this work is needed. As authors have explained in their response “high resolution, possibility of high magnifications, possibility of observing details not visible with other techniques such as light microscopy” , these type of discussion is needed in the intro to justify this work.

It is believed that the aim of the work it is not well defined and it is still ambiguous. Moreover, EDS is considered out of the scope of this work. It is believed that confuses the reader regarding the aim of the work which is not the documentation of the preservation state of the material.

Materials and Methods

Based on the fact that this work aimed to develop an SEM protocol mainly for wood ID it is considered that this section should be rearranged.

More specifically it is believed that material’s background info (2.1) and physical properties (2.3) should precede microscopy, so the reader will obtain a full picture of the condition of the wood examined before its microscopical investigation.

I understand that 2.3 has been relocated in order to fade its relative importance, however I feel that 2.3 section also provides background info and thus should appear before the principal section of the alternative SEM protocol.

A suggested structure is

2.1. Archaeological material..

2.2 Physical characterization….

2.3. SEM …, EDS…..

2.4 LM……

As it was mentioned earlier the section on EDS together with SEM is considered that diminish the value of the SEM protocol proposed. It is considered that EDS can be part of the physical characterization section (2.1) as it is an independent analysis of the inorganic elements of the material and is not related to wood morphological microscopic features

Regarding the light microscopy protocols that were applied comparatively are considered very important and this aspect of this work should more elaborated, mentioned and highlighted in both discussion and conclusions (pros and cons among SEM and LM protocols)

Results

Results presented did not prove that this alternative SEM protocol can be used for both wood identification and assessment of the state of wood preservation. The low magnification SEM micrographs presented (<x3000) and cross sections only, cannot demonstrate wood biodeterioration or other decay patterns justifying that the protocol has the potential to permit morphological assessment of highly degraded material. The whole section “Scanning electron microscopy” has not documented a single degradation pattern in order to justify the use of the protocol for assessing wood preservation state.

In contrast in the ID section the SEM micrographs, presented nicely, wood anatomical features showing that the protocol can be used for wood ID. Paradoxically, results on light microscopy protocols examined for comparative reasons, also showed that they can be well applied for wood ID. Thus, it is not either clear or justified whatsoever why the SEM should be preferred over LM.

The pros and cons between SEM and LM for wood ID are well recognized worldwide. If authors considered that SEM in terms of infrastructure, time, cost, easiness, applicability, reliability, is better than LM, it is going to be judged by the readers of the ms and those who will follow this alternative protocol. In my opinion someone with wood anatomy knowledge can ID a wood species having just one razor blade, some glycerin and a basic light microscope. Thus, it is considered that very strong scientific arguments are needed to support this statement, which were not provided herein.

Nonetheless for the scope of this review it is recommended to merge these two SEM sections and demonstrate the pros of this protocol in relation to wood ID  

Conclusions

As mentioned during the first round, the statement that the inclusions found in the wood indicate bacterial degradation of wood, cannot be supported by the obtained results and the statement is considered arbitrary. Which inorganic constituents are indicative of bacterial attack? Do you mean that sulfur reducing bacteria decay wood? Please clarify with relevant citations

Moreover, conclusion on the speed of analysis is also need to better clarified.  A large “2D” sample could have larger surface area and volume than a small 3D one.

Finally, major outcomes of this work are not considered that they have been well-mentioned, adressed  and highlighted appropriately i.e You have detected some wood features that could not be seen by other techniques because the wood structure is changed (decayed). This is very important! What features? This type of info is considered of great importance and must be in the concussions

Overall it is considered that this work could be really significant of content after revision. The potential of the alterative proposed SEM protocol should be focused on wood identification and the pros of this protocol should be emphasized.

If possible, results obtained by SEM for wood ID should be comparatively assessed to LM in order to demonstrate the proposed SEM protocol significance.

Beside these comments, some remarks and suggestions are annotated on the pdf of the revised manuscript.
